# Object Dynamics Distillation for Scene Decomposition and Representation

**Qu Tang** [12]    **XiangYu Zhu** [12]    **Zhen Lei** [123]    **ZhaoXiang Zhang** [123]∗

[1]School of Artificial Intelligence, University of Chinese Academy of Sciences
[2]Institute of Automation, Chinese Academy of Sciences
[3]Centre for Artificial Intelligence and Robotics, Hong Kong Institute of Science & Innovation,
Chinese Academy of Sciences
{tangqu2020,zhaoxiang.zhang}@ia.ac.cn, {xiangyu.zhu,zlei}@nlpr.ia.ac.cn

## Abstract

The ability to perceive scenes in terms of abstract entities is crucial for us to achieve higher-level intelligence. Recently, several methods have been proposed to learn object-centric representations of scenes with multiple objects, but most of them focus on static images. In this paper, we work on object dynamics and propose Object Dynamics Distillation Network (ODDN) which distills explicit object dynamic representations (e.g., velocity) from raw video input. Furthermore, we build a relation module that calculates object-pair interactions and applies it to the corresponding dynamic representations of objects. We verify our approach on tasks of video events reasoning and video prediction, which are two important evaluations for video understanding. The results show that visual representations of ODDN perform better in answering reasoning questions around physical events in a video compared to representations of the previous scene representation methods. And, ODDN could generate reasonable future frames given two input frames, considering occlusion and objects collision. In addition, using the object dynamic clues allows the model to obtain better scene decomposition quality in segmentation and reconstruction. Code is available at https://github.com/tqace/ODDN.

## 1 Introduction

Humans learn to decompose the scene into independent objects to understand the environment at a very early age. This cognitive pattern promotes the generation of common sense of physics, for example, the speed of the object will change after a collision. However, the neural networks seem to lack the ability to acquire a compositional understanding of the world in terms of structured objects, which is crucial for generalizing and higher-level tasks such as planning and reasoning(Greff et al., 2020). To tackle this problem, many scene decomposition approaches have been proposed to segment an image with ground-truth segmentation labels as supervision(Ronneberger et al., 2015; Long

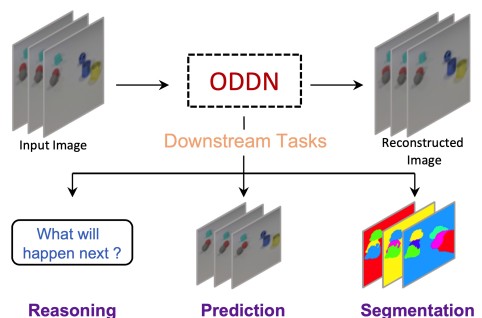

Figure 1: ODDN could be used to handle multiple downstream tasks.

et al., 2015; Badrinarayanan et al., 2017; Chen et al., 2017; Zhang et al., 2020a;b). However, the acquisition of semantic segmentation labels is costly. Besides, these methods do not provide structured representations of segmented areas. Recently, several unsupervised methods have become popular in learning decomposition and object-centric representations of scenes with multiple objects(Hsieh et al., 2018; Van Steenkiste et al., 2018; Burgess et al., 2019; Locatello et al., 2020; Greff et al.,

---

∗Corresponding author

2017; Eslami et al., 2016; Kosiorek et al., 2018), and some of those methods learn disentangled features which represent crucial properties (e.g., 'color', 'size') of object in the same format. The work of Van Steenkiste et al. (2019) has proved that disentangled representations do in fact lead to better downstream task performance.

Beyond static scenes (images), people understand dynamic physical events and predict how the future might unfold(Battaglia et al., 2013). Even infants have the intuition that temporarily occluded objects remain coherent wholes and follow spatially contiguous paths(Baillargeon, 1987). This common sense of physics is inseparable from the understanding of object dynamics. However, existing scene decomposition and representation work focus on images, which means that when interpreting videos, those approaches process one frame at a time, and do not encode objects' dynamic properties. We argue that representing object dynamics is significant for tasks such as video reasoning or predicting the future.

In this work, we aim to distill and disentangle object dynamics in an unsupervised manner and present ODDN (Object Dynamics Distillation Network). ODDN is built on a spatial mixtures framework(Greff et al., 2019), a prime example of VAE(Kingma & Welling, 2013) based architecture of decomposing scenes into object-centric latent representations. Formally, the model clusters and decodes pixels into latent representations through an iterative amortized variational inference procedure then reconstructs the latent vectors into input images. In this way, scenes are segmented and represented as object-centric disentangled features. Based on this, our key insight is that the object dynamics hide in variations between objects' static latent representations of different frames, thus, we design our model to distill object dynamics from their representations of the current and previous frames. With the distilled object dynamics as part of the representation, we further build a relation module to model object interactions.

We evaluate ODDN on different downstream tasks as shown in Figure 1. Results show that representations with object dynamics perform better in reasoning task than representations with static properties only, and the relation module endows ODDN with the capability of predicting the future. In addition, we notice that incorporating object dynamics and relations into the basic scene decomposition framework benefits the segmentation and reconstruction quality.

We highlight contributions of our approach:

- Distillation of disentangled object dynamics from raw video input in an unsupervised manner.
- State-of-the-art performance on CLEVRER(Yi et al., 2019) which focus on video understanding and causal reasoning.
- Capability of future frames prediction in 3D scenes including physical events (occlusion, collision).
- Better scene decomposition quality for spatial mixture architecture in terms of segmentation and reconstruction.

## 2 RELATED WORK

**Object representation of static scenes.** Representation learning is an important topic in deep learning. Several recent lines of work making breakthroughs in this area learns representations without explicit supervision. NEM(Greff et al., 2017) builds a spatial mixture model based on EM algorithm to discover a compositional object representation but fails to cope with colored or 3D data. AIR(Eslami et al., 2016) achieves scene interpretation by using a recurrent neural network that attends to one element at a time and disentangles positions and appearances explicitly in latent variables. AIR can decompose 3D scenes, however, it generates partial images separately and adds them together to form the final image, which leads to its restriction: it can't be applied to scenes with occluded objects. MoNet(Burgess et al., 2019) uses multiple attention steps to tackle complex 3D scene, which is the first being capable of discovering object masks and completing partially occluded objects. IODINE(Greff et al., 2019) does the same thing as MoNet by using iterative variational inference to refine the inferred latent representation in each encoding step.

**Object representation of dynamic scenes.** Methods mentioned above are designed to decompose static scenes, hence they do not encode object dynamics in the latent space. R-

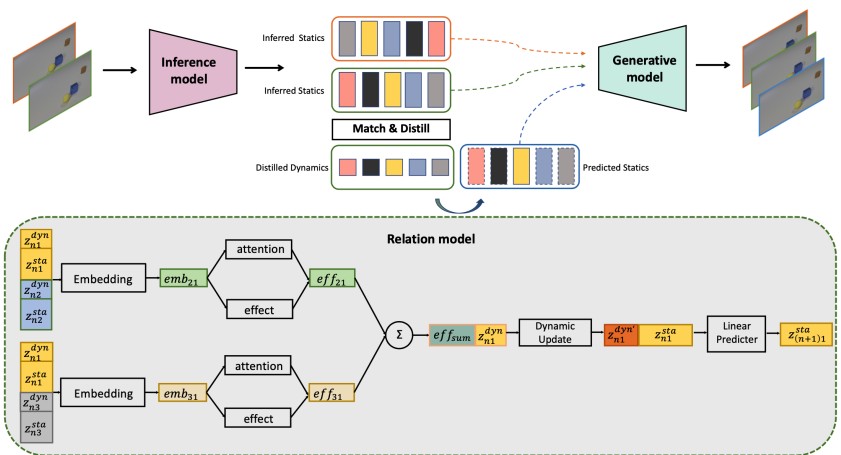

Figure 2: Overview of ODDN. The inference model assigns inputs into $K$ slots per image, each slot encodes object or background into latent space, shown in different colors. We take use of the attention mechanism of transformer to align objects' latents of two input images and encode the aggregated representation into a low-dimensional vector to obtain disentangled object dynamics. Object latents (sta) and dynamics (dyn) are used to model interactions between object-pairs, we show the relation module with three slots as an example, which compute the effects $z_2$ and $z_3$ have on $z_1$. We concat the updated object dynamics and object latents to predict the next frame object latents. The Generative model outputs both reconstructions of input images and prediction of future frame images.

NEM(Van Steenkiste et al., 2018) applies RNN to NEM to encode object dynamics implicitly, and models relations between objects to learn physical interaction which enables the model to predict future frames for video input. Similarly, SQAIR(Kosiorek et al., 2018) extends AIR to image sequence to discover and track objects throughout the video frames. Both R-NEM and SQAIR bear the drawbacks of their predecessors. ViMON(Weis et al., 2020) is a video-extension of MoNet by building temporal connections between latent representations with GRU(Cho et al., 2014). ViMON learns a linear transformation to predict next time-step image, encouraging dynamics encoded. PRO-VIDE(Zablotskaia et al., 2020) applies 2D-LSTM(Graves et al., 2007) to IODINE and is capable of simulating future video frames. Although ViMON and PROVIDE encode object dynamics, they do not model interactions between objects, thus, they can't handle datasets with object collisions. Besides, object dynamics of both methods are spread in latent space implicitly, which lacks interpretability. It has been proved that disentangled representations allow one to learn models with better performance and fewer samples for downstream reasoning tasks(Wu et al., 2021; Chen et al., 2021b; Gan et al., 2017; Ding et al., 2021b). Therefore, in this work, we aim to represent object dynamics explicitly. Instead of using RNNs, we match objects' latent representations between the current and previous frames and distill their dynamics by applying a single-layer transformer encoder.

## 3 METHOD

Our goal is to decompose dynamic scenes into object-centric representations purely based on visual observations. And the representation contains interpretable object dynamics. Recent works decompose static scenes by randomly assigning objects to a fixed number of slots, which share weights to obtain latent vectors with a common format. However, these works do not encode object dynamics for objects of sequential input. Our key idea is that building upon these successful multi-object scenes decomposing and representation approaches, object dynamics could be distilled from the corresponding latent representations of two consecutive frames.

### 3.1 OBJECT DYNAMICS DISTILLATION NETWORK

We illustrate the framework of ODDN in Figure 2. Formally, ODDN takes as input a video, then the inference model processes all frames $x_{1:N}$ independently to obtain $K$ object latent vectors which encode object static properties. For frames $x_{2:N}$ , we distill objects' dynamics from their latent

vectors of the current and the previous frames. The relation module takes the object latent vectors and the object dynamics of frames $x_{2:N-1}$ as input, calculates the interaction between objects and predicts object latent representations of frames $x_{3:N}$.

The input image $x \in \mathbb{R}^D$ could be reconstructed by $K$ latent vectors $z_k \in \mathbb{R}^M$ , each of which corresponds to one object or background area after training, and each latent vector is also called a slot. Firstly, latent $z_k$ is decoded into pixel-wise mean $\mu_{ik}$ and pixel-wise mask logit $\hat{m}_{ik}$ . Then, the $K$ mask logits are normalized with a softmax function to ensure that the sum of $K$ mask logits for each pixel is 1. At last, $\mu$ and $m$ parameterize the final spatial Gaussian mixture distribution:

$$p(x|z) = \prod_{i=1}^{D} \sum_{k=1}^{K} m_{ik} \mathcal{N}(x_i; \mu_{ik}, \sigma^2), \tag{1}$$

where $\sigma$ is the same and fixed for all $i$ and $k$. The input image can be reconstructed as $\hat{x} = \sum_{k=1}^{K} m_k \mu_k$.

The inference model of ODDN follows previous work (Greff et al., 2019) by using amortized variational inference to infer and refine posterior $q_\lambda(z|x)$ iteratively where $\lambda = \{\mu_z, \sigma_z\}$. The main reason to use such an iterative process is that the standard variational inference framework leads to the multi-modal problem, which means that any pixel could be captured by any slot. IODINE updates posterior as follows:

$$z^t \sim q_\lambda(z^t|x), \tag{2}$$

$$\lambda^{t+1} \leftarrow \lambda^t + f_\phi(z^t, x, a), \tag{3}$$

where the posterior parameters $\lambda$ start with an arbitrary guess, and the first latent representations $z^1$ are sampled from $q_\lambda$. In each refinement step, input $x$ together with a set of variables $a$(means $\mu$, masks $m$, mean gradient $\nabla_\mu \mathcal{L}$, mask gradient $\nabla_m \mathcal{L}$), which are computed from $z^t$ are fed into a refinement network $f_\phi$ to obtain the additive updates for the posterior.

ODDN predicts future objects state in latent space given first two frames:

$$\hat{z}_{nk} = W_{pred}([z_{(n-1)k}^{dyn}; z_{(n-1)k}^{sta}]); \quad n > 2, \tag{4}$$

where $z^{dyn}$ is the distilled object dynamic representation and $z^{sta}$ refers to the object latent representation output by the inference model. The next-frame latent vectors are predicted with a learnable transformation $W_{pred}$, details will be introduced in Section 3.2 and Section 3.3. The whole network is trained end-to-end and the final loss consists of prediction loss $\mathcal{L}_p$ and reconstruction loss $\mathcal{L}_r$, in detail:

$$\mathcal{L}_r(x) = \sum_{n=1}^{N} \sum_{t=1}^{T} \left[ -log(p(x_n|z_{n\cdot}^t)) + \beta KL(q_\lambda(z_{n\cdot}^t|x_n, z_{n\cdot}^{<t})||p(z)) \right], \tag{5}$$

$$\mathcal{L}_p(x) = \sum_{n=3}^{N} \left[ -log(p(x_n|\hat{z}_n)) \right], \tag{6}$$

where $T$ is the maximum refinement step number during the inference stage. We measure and optimize the KL divergence between the variational posterior and the prior in $\mathcal{L}r$.

## 3.2 DYNAMIC DISTILLATION

In this section, we describe how ODDN derive object dynamics from sequential object latent representations. For existing scene decomposition and representation approaches, object-to-slot assignments usually switch unpredictably in a video. Thus, we draw support from the Transformer to match object latent representations in different frames correctly and further distill object dynamics.

**Revisit attention mechanism of Transformer.** The key component of the Transformer is the attention module which could be mathematically described as:

$$Attention(q, K, V) = \sum_{i=1}^{|V|} \alpha_i W_v v_i; \quad \alpha_i \propto exp\left[ (W_q q)^T (W_k k_i) \right], \tag{7}$$

where $q \in \mathbb{R}_d$; $K$ consists of $|K|$ elements $(k_1, k_2, \cdots, k_{|K|})$ with $k_i \in \mathbb{R}_d$; $V$ consists of $|V|$ elements $(v_1, v_2, \cdots, v_{|V|})$ with $v_i \in \mathbb{R}_d$.

This attention function maps the input query $q$ and a set of key($k$)-value($v$) pairs to an updated representation, $W_q$, $W_k$, $W_v$ are learnable parameters. Besides, $\alpha_i \geq 0$, $\sum_{i=1}^{|v|} \alpha_i = 1$, and usually $|K| = |V|$.

**Model architecture.** The inference model decomposes the input video into $N \times K$ latent vectors, where $K$ is the slot number for each frame and $N$ is the number of frames of the video. We put these sequence consisting latent vectors $\{z_{11}, \cdots, z_{1K}, \cdots, z_{N1}, \cdots, z_{NK}\}$ into one single Transformer encoder layer to match the corresponding slots which usually switch their assignments with objects several times in a video. After finding object correspondence in different frames, the Transformer aggregates latent information and distills object dynamics which are encoded into a low-dimensional representation space. Mathematically, the dynamic distillation process for the $k$-th slot in the $n$-th frame could be described as:

$$z'_{nk} = Attention(z_{nk}, \{z_{n-1}, z_n\}, \{z_{n-1}, z_n\}); \quad n > 1, \tag{8}$$

$$z_{nk}^{dyn} = FFN(Norm(z'_{nk} + z_{nk})), \tag{9}$$

where $z^{dyn}$ is a low-dimensional vector transformed by the fully connected feed-forward network (FFN). This learning process is performed in parallel by pre-defining an attention mask to ensure that latent representations of the current frame only attend to ones one frame before.

## 3.3 RELATION MODULE

ODDN models interactions between object pairs by computing all object-pair effects and letting them act on and update object dynamics. ODDN learns a soft attention coefficient for each effect. The object dynamics are updated as:

$$emb_{k_i,k_j} = W_{emb}([z_{k_i}^{dyn}; z_{k_i}^{sta}; z_{k_j}^{dyn}; z_{k_j}^{sta}]), \tag{10}$$

$$att_{k_i,k_j} = Sigmoid(W_{attn}(emb_{k_i,k_j})), \tag{11}$$

$$eff_{k_i,k_j} = W_{eff}(emb_{k_i,k_j}), E_{k_i} = \sum_{i \neq j} att_{k_i,k_j} * eff_{k_i,k_j}, \tag{12}$$

$$z_{nk_i}^{dyn'} = z_{nk_i}^{dyn} + W_{dyn}(E_{k_i}), \tag{13}$$

where $W_{emb}$ first embeds representations of object-pair, then $W_{att}$ learns the attention coefficient based on the object-pair embedding, and $W_{eff}$ learns the effect $z_{k_j}$ has on $z_{k_i}$. At last, the summed effects output an update value with $W_{dyn}$.

We draw inspiration from NPE(Chang et al., 2016) and R-NEM(Van Steenkiste et al., 2018), both architectures model object relations. NPE has access to ground truth representations of scenes and computes effect for each neighborhood object-pair with a neighborhood mask based on ground-truth object position, and finally, the summed effects act on object velocity. R-NEM represents objects in an unsupervised fashion and implicitly encodes sequential information with an RNN(Hochreiter & Schmidhuber, 1997). Instead of using a neighborhood mask, R-NEM learns soft attention for all object-pairs. Our approach learns to represent object dynamics explicitly. Thus, we compute all object-pair effects and let it act on object dynamics as NPE, and learn an attention coefficient for each effect as R-NEM.

After distilling and updating object dynamics, we predict next-frame object latents by simply learning a transformation $W_{pred}$, and $W_{pred} \in \mathbb{R}^{(L+l) \times L}$, $\hat{z}_{(n+1).}^{sta} = W_{pred}([z_{n.}^{dyn'}; z_{n.}^{sta}])$, where $l$ indicates the dynamics dimension ($l = 4$ in our experiments).

To alleviate error accumulating when implementing multi-step predicting in the inference stage, we use student forcing strategy during training: instead of only using $z_n$ to predict $\hat{z}_{n+1}$, $\hat{z}_n$ is also used to iteratively roll out a predicted image sequence.

Table 1: Performance (per question/option accuracy) comparison with the state-of-the-art methods MAC (V+), NS-DR, DCL, and ALOE on CLEVRER, ALOE takes static scene representations from Monet as visual input while ODDN-ALOE use representations of ODDN.

| Model | Descriptive | Explanatory | | Predictive | | Counterfactual | |
|---|---|---|---|---|---|---|---|
| | | per opt. | per ques. | per opt. | per ques. | per opt. | per ques. |
| MAC(V+)(Yi et al., 2019) | 86.4 | 70.5 | 22.3 | 59.7 | 42.9 | 63.5 | 25.1 |
| NS-DR(Yi et al., 2019) | 88.1 | 87.6 | 79.6 | 82.9 | 68.7 | 74.1 | 42.2 |
| DCL(Chen et al., 2021a) | 90.7 | 89.6 | 82.8 | 90.5 | 82.0 | 80.4 | 46.5 |
| ALOE(Ding et al., 2021a) | 94.0 | 98.5 | 96.0 | 93.5 | 87.5 | 91.4 | 75.6 |
| ODDN-ALOE | **95.8** | **98.9** | **97.0** | **95.7** | **91.8** | **93.0** | **80.1** |

Table 2: Ablations of how different object representations perform on CLEVRER, all experiments are conducted based on ALOE without self-supervision. "ODDN w/o relation" is ODDN trained without relation module (with only dynamic distillation module). $-dyn$ means reducing distilled dynamic representations (only static representations remained). "MoNet w/o ss" is the result of ALOE without self-supervision from original paper with MoNet features.

| Feature | Dim | Dyn | Descriptive | Explanatory | | Predictive | | Counterfactual | |
|---|---|---|---|---|---|---|---|---|---|
| | | | | per opt. | per ques. | per opt. | per ques. | per opt. | per ques. |
| PROVIDE | 16 | ✓ | 75.2 | 91.6 | 77.4 | 80.5 | 65.4 | 76.4 | 39.4 |
| MoNet w/o ss | 16 | | 91.0 | - | 92.8 | - | 82.8 | - | 68.7 |
| IODINE | 20 | | 92.8 | 98.3 | 95.6 | 89.1 | 80.2 | 89.7 | 71.3 |
| IODINE | 16 | | 93.5 | 98.1 | 95.0 | 91.6 | 84.5 | 90.6 | 73.4 |
| ODDN w/o relation$^{(-dyn)}$ | 16 | | 92.8 | 97.6 | 93.6 | 90.6 | 82.8 | 89.2 | 70.1 |
| ODDN w/o relation | 20 | ✓ | 93.9 | 98.2 | 95.1 | 93.6 | 88.0 | 90.7 | 73.6 |
| ODDN$^{(-dyn)}$ | 16 | | 95.1 | 98.5 | 96.1 | 94.1 | 89.1 | 91.7 | 76.8 |
| ODDN | 20 | ✓ | **95.8** | **98.9** | **97.0** | **95.7** | **91.8** | **93.0** | **80.1** |

## 4 EXPERIMENTS

In this section, we design experiments in the perspective of representation, prediction, and scene decomposition. In particular, we study how ODDN performs on tasks of video understanding and reasoning, video prediction, reconstruction, and segmentation. To quantify segmentation quality, we measure the similarity between ground-truth (instance) segmentations and our predicted object masks using the Adjusted Rand Index(Rand, 1971) (ARI) and The Foreground Adjusted Rand Index (F-ARI) which is a modification of the ARI score ignoring background pixels.

**Baselines.** We compare our method with IODINE, PROVIDE and SRVP(Franceschi et al., 2020). IODINE is designed for static images, thus it neither encodes object dynamics nor considers object relations. PROVIDE is an extension of IODINE to fit sequential inputs. The major difference between PROVIDE and our approach is that PROVIDE encodes object dynamics implicitly while we distill object dynamics explicitly. In detail, PROVIDE uses 2D-LSTM to achieve temporally conditioned and iterative amortized inference for posterior refinement, as a result, the final sampled latents in every time-step contain dynamic information which could be used to predict the future. Besides, PROVIDE does not model object relations, which means when predicting the future, object latents are processed separately. SRVP is a dynamic latent model for stochastic video prediction which decouples frame synthesis and dynamics based on residual updates of a small latent state. Unlike ODDN and PROVIDE, SRVP reperesents the scene with a single latent.

### 4.1 VIDEO UNDERSTANDING AND REASONING

We study the representation power of ODDN on CLEVRER, which poses different types of questions for each video in CLEVRER dataset. Those questions could test the model's understanding of a video comprehensively, including descriptive questions ("how many objects are moving"), explanatory questions("the reason why certain collision happens"), predictive questions("what will happen next"), and counterfactual questions("what would happen if").

Table 1 shows that ODDN-ALOE outperforms ALOE on all question types even without self-supervision, especially on predictive and counterfactual questions, and achieves state-of-the-art performance. Both predictive and counterfactual types of questions are related to predicting how objects

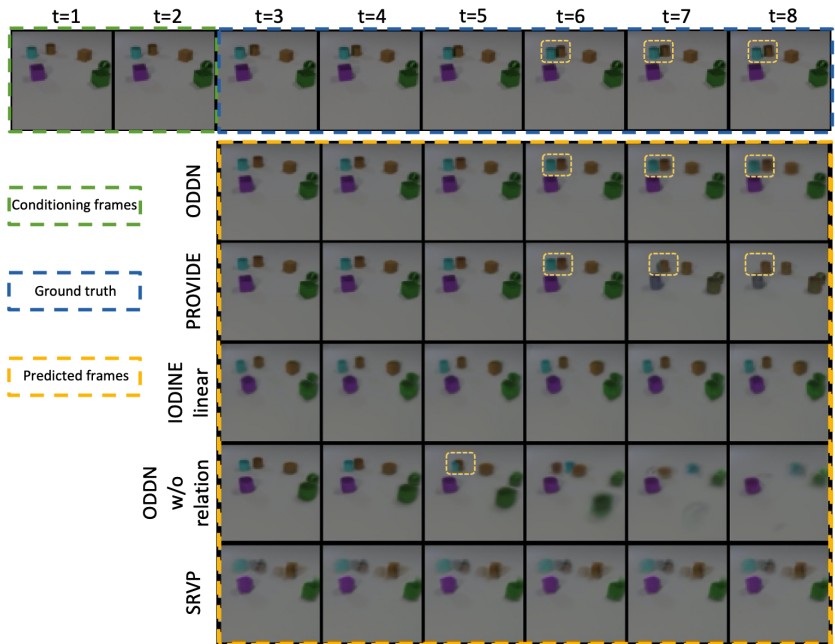

Figure 3: Illustration of video prediction results on CLEVRER. All models take as input the first two frames and predict the next six frames. "IODINE linear" is implemented by simply concatenating latent features of the current and the previous frames and predicting the future frame with a feedforward neural network. "ODDN w/o relation" is the version of ODDN without relation module. In the video, a collision happens between the blue and brown cylinder from time step 6 to 8.

will move. This result proves that representations with object dynamics could help the neural model to do better understanding and reasoning around physical events. We also do ablation experiments to study how different representations affect the model's performance, results are shown in Table 2. We can tell that representations of PROVIDE lead to a significant decline compared to IODINE, and the reason will be discussed in Section 4.3. On the contrary, representations of ODDN which encode object dynamics (dim=20) outperform IODINE (dim=16) by a large margin, especially on predictive (91.8 vs 84.5) and counterfactual (80.1 vs 73.4) questions. In addition, we obtain ODDN features without dynamics by reducing 4 dimension dynamics. Results show that ODDN without dynamics still performs better than IODINE, which means that ODDN not only distills useful object dynamics but also generates better static representations. static representations. "ODDN w/o relation" outperforms "ODDN w/o relation $-dyn$" which further shows the significance of our distilled object dynamics. However, "DDN w/o relation $-dyn$" is no better than IODINE (dim=16), which proves that the relation module is the main reason for the quality increase of the object static representation.

## 4.2 PREDICTION

ODDN achieves predicting future frames in an iterative fashion. Unlike PROVIDE, ODDN models object dynamics and interactions explicitly. Results could be observed in Figure 3, where the blue and brown cylinder collides. ODDN could output reasonable predictions and handle objects collision. PROVIDE could only predict object's future state independently, and the reason is that PROVIDE does not cope with objects interaction, thus, objects blend when they suppose to collide and change their dynamics. ODDN also generates more stable predictions comparing to PROVIDE which fails to capture the object's correct properties (shape, color) after a few frames. "IODINE linear" and SRVP generate identical pictures, which means that both methods are incapable of learning object-level dynamics on CLEVRER. "ODDN w/o relation" learn some dynamics, but the generated results are not reasonable, which may be caused by the training noise: predicting the collision without modeling object interactions. We calculated the average MSE of each of the predicted 6 frames over the entire test set, the quantitative results are shown in Figure 7.

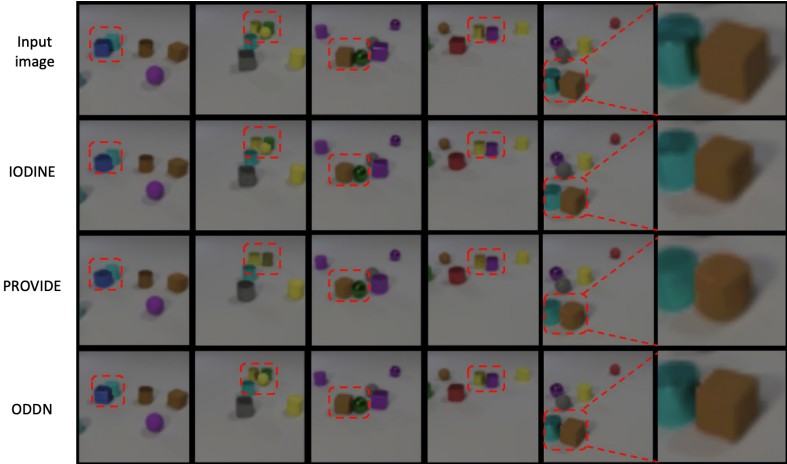

Figure 4: Comparison of reconstruction results of ODDN with IODINE and PROVIDE, we highlight dense areas with dashed boxes.

## 4.3 DECOMPOSITION

**Reconstruction.** We show how different architectures perform in terms of reconstruction quality in Table 3. Comparing to IODINE, PROVIDE brings some negative effects, as shown in Figure 4. PROVIDE occasionally fails to capture the right object shape. Instead, we notice that ODDN performs better than IODINE. Specifically, the reconstruction of close objects is more precise. We believe that this benefits from the relation module which relies on object surface details to process objects interaction (whether collision occurs and how object dynamics update). The reconstruction performance is positively correlated to object representation quality which has been verified in Section 4.1.

**Segmentation.** ODDN and baseline models decode object latents into pixel values and segmentation masks. The major contribution of PROVIDE is that it models temporal dependencies between latents across frames to obtain better segmentation results.

Table 3 indicates that PROVIDE hurts the model's reconstruction capability. Instead,

Table 3: We compare ODDN with baseline models on scene decomposition quality, ODDN-$l$ indicates training with the long-range prediction (6 frames).

| Model | Dyn | MSE ↓ | ARI↑ | F-ARI ↑ |
|---|---|---|---|---|
| IODINE | | 0.000100 | 0.096 | 95.55 |
| PROVIDE | ✓ | 0.000148 | 0.228 | 95.17 |
| ODDN | ✓ | **0.000075** | 0.156 | **95.63** |
| ODDN-$l$ | ✓ | 0.000103 | 0.233 | 95.36 |

we show that the future frame prediction task in our architecture also improves the segmentation performance and achieves good reconstruction quality at the same time.

The ARI further improves when increasing the steps of future object states to predict. We believe that the long-range prediction supervision encourages ODDN to attend more on foreground objects which may change its position and view over time. This temporal consistency is similar to PROVIDE, which could be the main reason for both models to achieve better segmentation performance than IODINE. Results are shown in Figure 5.

## 4.4 REAL BLOCK TOWERS

**Setup.** We further evaluate ODDN on the real block tower video dataset(Lerer et al., 2016). The block tower dataset has 493 real-world videos and each video contains a block tower which may or may not be falling. We train models on 393 randomly-selected videos and evaluate them on the rest 100 videos. We study how the models perform in reconstruction and predicting the future frames with the physical concept of "falling".

**Results.** We show reconstruction results by ODDN and baseline models of images with block towers of different inclination in Figure 6. We found that all models can reconstruct images of stable block towers precisely. However, for blocks with complex spatial position states (e.g., falling

with interactions), IODINE and PROVIDE often fail to restore accurate block properties. ODDN performs the best and is able to retain the shape and inclination angle of the object.

In Figure 8, we present prediction results of ODDN, PROVIDE and SRVP. SRVP predicts reasonable object dynamics but is not accurate, and the generated images are not clear. The blocks predicted by PROVIDE tend to be blurred and fused over time. ODDN models object interactions, thus predicted blocks do not blend easily.

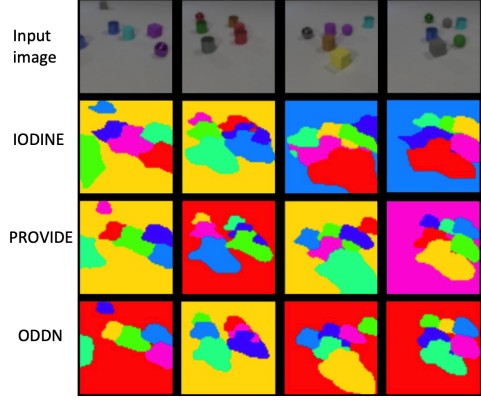

Figure 5: Unsupervised Segmentation results of ODDN, PROVIDE, and IODINE on CLEVRER. Each color corresponds to one slot index whose assignment is unpredictable. Both PROVIDE and ODDN generate better segmentation masks than IODINE.

Figure 6: Qualitative comparisons of reconstruction of ODDN with IODINE and with PROVIDE on realistic block towers.

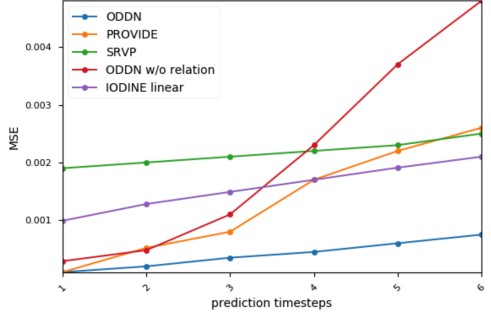

Figure 7: Comparison of MSE for 6 predicted frames of different models on CLEVRER.

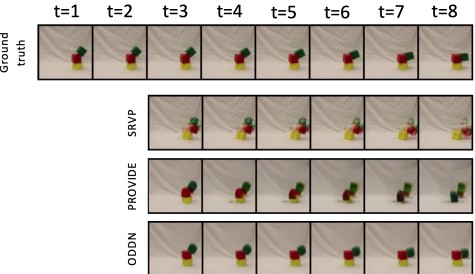

Figure 8: Qualitative comparison of prediction performance of ODDN with PROVIDE and SRVP on realistic block towers.

## 5 DISCUSSION AND FUTURE WORK

We presented an unsupervised learning framework ODDN, a novel approach to decompose temporal scene of multiple objects with dynamics and interactions. ODDN distills disentangled object dynamics which are significant in downstream video reasoning task. ODDN also models object interactions which endows the model with the capability of predicting future frames. The incorporating of object dynamics and interactions benefits the model's decomposing ability, reflecting in better segmentation mask and reconstruction quality.

ODDN also has limitations. The prediction quality decrease over time because of the error accumulation. ODDN can only generate deterministic predictions for dynamics obeying regular patterns. Besides, ODDN needs two frames to firstly derive object dynamics then predict the future. However, in the block tower case, humans can tell whether the tower will fall or stay stable with one frame by commonsense. We leave this for future work.

ACKNOWLEDGMENTS

This work was supported in part by the National Key Research Development Program (No. 2020YFC2003901), Chinese National Natural Science Foundation Projects 62176256, 61876178, 61976229, the Youth Innovation Promotion Association CAS (Y2021131), and the InnoHK program.

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

## A  DETAILS ON METHOD AND IMPLEMENTATION

### A.1  ALGORITHM

Here we detail our algorithm with pseudocode:

---

**Algorithm 1:** ODDN Pseudocode

---

**Input:** video $x_1 \cdots x_N$, hyperparameters $K, T$

**Input:** trainable parameters $\lambda^{(1)}, \theta, \phi, \rho, \delta, \sigma$

1 **Function** `InferReconstruct(x):`

2      **for** $t = 1$ **to** $T$ **do**

3          $z_k^{(t)} \sim q_\lambda(z_k^{(t)}|x)$

4          $\mu_k^{(t)}, \hat{m}_k^{(t)} \leftarrow Decoder_\theta(z_k^{(t)})$

5          $m(t) \leftarrow softmax(\hat{m}_k^{(t)})$

6          $p(x|z^{(t)}) \leftarrow \sum_k m_k^{(t)} \mathcal{N}(x; \mu_k^{(t)}, \sigma^2)$

7          $\mathcal{L}_r^{(t)}(x) \leftarrow D_{KL}(q_\lambda(z^{(t)}|x)||p(z)) - \beta log(p(x|z^{(t)}))$

8          $a_k \leftarrow (x, z_k^{(t)}, \lambda_k^{(t)})$

9          $\lambda_k^{(t+1)}, h(t+1) \leftarrow f_\phi(a_k, h_k^{(t)})$

10      **end for**

11      **return** $\mathcal{L}_r, z^{(T)}$

12 **for** $n = 1$ **to** $N - 2$ **do**

13      $z_{n\cdot}, \mathcal{L}_r(x_n) \leftarrow InferReconstruct(x_n)$

14      $z_{(n+1)\cdot}, \mathcal{L}_r(x_{(n+1)}) \leftarrow InferReconstruct(x_{n+1})$

15      $dyn_{(n+1)\cdot} \leftarrow Transformer_\rho(z_{n\cdot}, z_{(n+1)\cdot})$

16      $dyn'_{(n+1)\cdot} \leftarrow Relation_\delta(dyn_{(n+1)\cdot}, z_{(n+1)\cdot})$

17      $\hat{z}_{(n+2)k} \leftarrow Predictor_\sigma(dyn'_{(n+1)k}, z_{(n+1)k})$

18      $\mathcal{L}_p(x_{(n+2)}) \leftarrow -log(p(x_{(n+2)}|\hat{z}_{(n+2)}))$

**Output:** $\mathcal{L}_p + \mathcal{L}_r$

---

### A.2  DATASET

CLEVRER is a synthetic video dataset of moving and colliding objects. Each video contains 128 frames at resolution $480 \times 320$. We pre-train ODDN on the entire CLEVRER training set, in order to promote convergence of our Relation Module, we extract images every 4 frames and ensure that at least one collision event is included forming CLEVRER-collision and we fine-tune ODDN on CLEVRER-collision. For testing, we use the validation set which has ground truth masks, we sample 1k sub-clips containing 6 objects, each sub-clip consists of 10 frames.

### A.3  HYPER-PARAMETERS

We Generally follow the setting of PROVIDE. We initialize the parameters of the posterior  by sampling from U(0.5,0.5). In experiments in prediction tasks, we use a latent dimensionality of 64 and downscale the image into $64 \times 64$ after a center-crop preprocess as IODINE and PROVIDE, such that dim($\lambda$) = 128. And in experiments in video reasoning task, we use a latent dimensionality of

Table 4: Computational complexity comparison between ODDN and IODINE and PROVIDE.

| Model | GFLOPs | Parameters | Model Size |
|---|---|---|---|
| IODINE | 17.2187 | 2780052 | 10.61 |
| PROVIDE | 17.2251 | 2977940 | 11.36 |
| ODDN | 17.2189 | 2925365 | 11.16 |

16 as ALOE which makes $\dim(\lambda) = 32$, and downscale the image into $64 \times 96$ without crop. The variance of the likelihood is set to $\sigma = 0.3$ in all experiments. We keep the default number of iterative refinements at $R = 5$, and use $K = 7$ slots for both training and testing. Furthermore, we set $\beta = 100.0$ for all experiments.

### A.4 TRAINING

We train our models on 8 GeForce RTX 3090 GPUs, which takes approximately two days per model. We use ADAM for all experiments, with a learning rate of 0.0003 and default values for all remaining parameters. During training, we gradually increase the number of frames per video, as we have found this to make the optimisation more stable. We train models with sequences of length 8 and the batch size is 12.

## B  MODEL COMPUTATIONAL COMPLEXITY

## C  MORE VISUALIZATION RESULTS

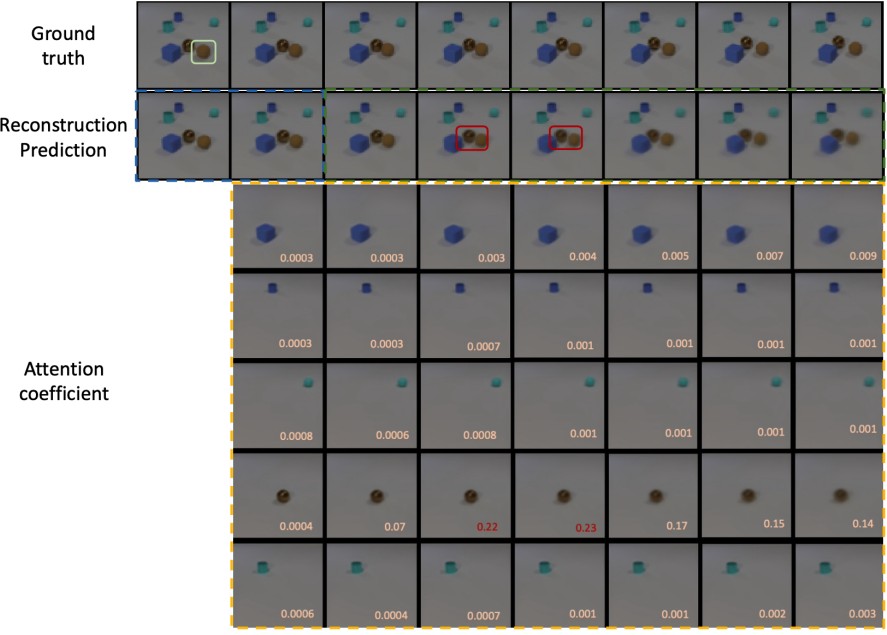

Figure 9: Addition prediction results of ODDN on CLEVRER and detailed attention coefficients visualization for relation module. We plot attention the scores all objects have on the rubber sphere from the second frame. The collision happens between the rubber sphere and the metal sphere.

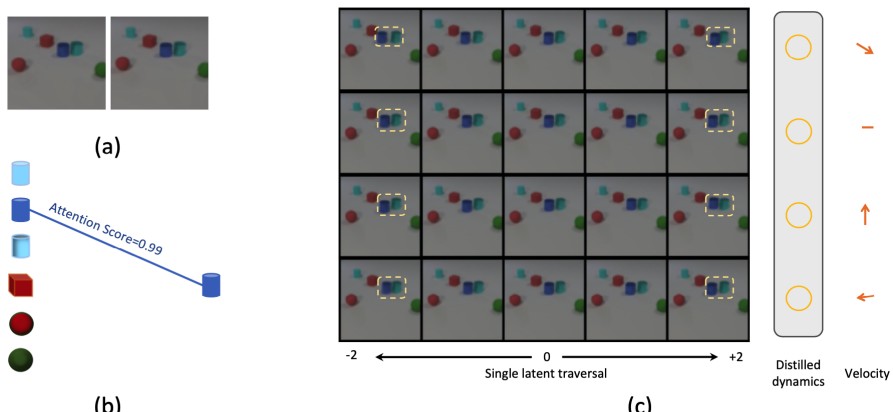

Figure 10: (a) Two consecutive frames with only the blue cylinder moving. (b) Visualization of the Transformer attention map of the blue cylinder in the matching phase of dynamics distillation. Latents of the second frame will update its representations by attending and merge information from latents of the first frame. Line intensity indicates the magnitude of attention probability. (c) Disentanglement visualization of the learned object dynamics. We adjust the feature value (from -2 to 2) of the corresponding dynamic representation dimension of the blue cylinder, and visualize the effects on the predicted next frame. We use arrows to show the velocity direction each dimension represents.

