# OpenReview forum: "OBJECT DYNAMICS DISTILLATION FOR SCENE DECOMPOSITION AND REPRESENTATION"
_ICLR.cc/2022/Conference — ICLR 2022 Poster_

### Official Review · Reviewer_YU79 · 2021-11-02

**Correctness:** 3
**Technical Novelty And Significance:** 3
**Empirical Novelty And Significance:** 2
**Recommendation:** 6
**Confidence:** 3

**Main Review:**

Strength:
+ The introduced modules, dynamic distillation and relation modules, are well motivated and their design and realization seem reasonable.
+ The paper reports strong numbers on the task of video reasoning outperforming existing works on CLEVRER.

Weaknesses:
- Missing ablation experiment for dynamic distillation and relation module. The paper claims that the slot assignment switches unpredictably inside the video. It proposed the dynamic distillation attention mechanism to solve that. Yet, it is not evaluated if this module actually solves this problem. With that, the reader does not know if this module is effective. The same for the relation module. The paper should report numbers with and without the use of this module to assess its impact.
- Missing baseline. To assess the significance of the proposed modules (dynamic distillation + relation module), the reader also needs results without them. This baseline can be created by replacing the proposed modules with a feed forward neural network (with roughly the same number of parameters as the proposed modules) that predicts the next static representation (t+1) based on the concatenation of the (static) representation at time t and t-1. The paper introduces some kind of baseline in Table 2 (ODDN w/o dyn) by reducing the 4 dimension dynamics. Thus, the network is left to predict the next state solely based on the current static representation (at time t) which is insufficient. That is why I urge the authors to implement the baseline as pointed out above and to report the resulting numbers in the rebuttal.
- The paper compares their method to PROVIDE which is the only other competitor that leverages temporal information. PROVIDE extends upon the static method of IODINE using a 2D-LSTM. The reported numbers in table 2 for PROVIDE are significantly (~20%) worse than for IODINE which does not make sense to me, especially when looking at the PROVIDE paper in which PROVIDE is on par with IODINE (on average). The same holds true for table 3. The MSE reported in PROVIDE is significantly better than IODINE whereas in this paper MSE is significantly worse (48% for MSE). This suggests to me that there is something wrong with how PROVIDE was trained/evaluated. The paper discusses a reason why ODDN is superior compared to PROVIDE in 4.3 but does not address why there is such a huge gap between IODINE and PROVIDE. Could the authors please clarify this for me? This shows and enhances the need for a baseline as discussed above.
- Missing implementation details. The paper does not provide any implementation details like e.g. optimizer, learning rate, \beta value (hyperparameters, etc). This makes the paper not reproducible and I urge the authors to add these details. Does the code get released after acceptance?
- The paper evaluates their proposed model only on the CLEVRER dataset. To better assess the impact of this work the method needs to be evaluated at least on one other dataset such as e.g. CATER.
CATER: Rohit Girdhar et al. CATER: A diagnostic dataset for Compositional Actions and Temporal Reasoning. ICLR 2020
-Missing related work. This paper misses a relevant related work from Veerapaneni et al. “Entity Abstraction in Visual Model-Based Reinforcement Learning” (CoRL 2019). This work extracts entity representations from images and learns their dynamics for model-based reinforcement learning.  Moreover, this paper only cites related work which operates on multi-object representations. Yet, there exists related work also for non-multi-object dynamics representation learning which should be cited such as: \
 Minderer et al. “Unsupervised learning of object structure and dynamics from videos” NeurIPS 2019 \
 Blattmann et al. “Understanding Object Dynamics for Interactive Image-to-Video Synthesis” CVPR 2021 \
Also, the work should cite major works in video prediction since it is the task the network gets trained on: \
Lee et al. “Stochastic Adversarial Video Prediction” (SAVP) \
Franceschi et al. “Stochastic Latent Residual Video Prediction” (SLRVP) \
Denton et al. “Stochastic video generation with a learned prior” (SVG)
- Missing comparison to current non-multi object representation-based approaches for video prediction. It would have been nice if the paper would have compared to non-multi object video prediction approaches to put their work more into context.
- This work does not deal with the inherent ambiguity of the video prediction task itself. It deterministically predicts a next state given two past observations. Thereby, it cannot cover the different scenarios the future can hold. The work from Lee et al. actually showed that by accounting for this ambiguity, the learned model improves on the task of video prediction (since it does not average over multiple predictions) and can cover multiple different future scenarios.  I would like to hear the authors take on that and see this more as future work for multi-object dynamics learning.

General notes:
- Below (4) there is z^{dyn} twice I believe the second one should be z^{sta}
- Paper states that their approach is capable of future frames prediction in “complex” 3D scenes. I have to disagree with this since this dataset on which the method is evaluated is far away from any real-world video dataset that is why I would not call it “complex”.
- Change the name from “Reported” to “ALOE w/o self-superv.” to make it easier to read in table 2.
typos: figure 7 caption it’s -> its; very young infants -> infants; line intensity -> Line intensity...
- Figure 7 has a,b and c but in the caption only a and b occur. In c) the x-axis goes from -2 to 2 but there are 10 images -> -3 to 3?
- The paper mentioned that the “Object’s motions … has to be *continuous* through space” while in the next sentence explaining how the next *discrete* static object latent state is predicted. One would have to use e.g. Neural ODEs to actually model continuous object motion so continuous should be replaced by e.g. smooth.
- The writing of the paper needs some polishing since the wording in many sentences does not make sense e.g. “Object’s motion consists with the basic physics concept ...”.
- In the caption of 7b it says: “line intensity indicates the magnitude of attention probability” while in the figure it is written that the score of blue is 0.99. To me, all the intensities look more or less the same while one of them is already 0.99 out of 1 (the other ones should be then ~0.01). How does that make sense?  Could the authors not show all the weights between all objects between two frames (also for multiple examples). This would provide much more insight to the reader.
- It is difficult to spot the object movement in figure 7 c. It would be great if the authors could add some markers to help visibility.



**Summary Of The Paper:**

This paper aims at learning object dynamics from unlabelled videos for multi-object representations. This work builds upon the inference model from Greff et al. (2019) to obtain the multi-object representations. Using that, this paper learns latent dynamics by predicting the future frame given two previous observations. The proposed network is trained by maximizing the log-likelihood in the pixel space for reconstructions as well as for predictions, similar to standard work in video prediction. In addition to that, this paper optimizes the objective from Greff et al. (2019) to obtain multi-object representations. The main contribution of this work lies in the module which predicts the future multi-object representation given the two past ones by using i) dynamic distillation as well as ii) a relation module.
The dynamic distillation module is motivated by the claim that the object-to-slot assignment switches inside videos. Therefore, the paper introduces a self-attention layer to match the slots of the specific objects across two frames. The output of this module after applying the feed-forward network is called dynamic representation. \
The relation module is proposed to model the interactions between object pairs. This is implemented (10-13) using self-attention between the dynamics (from above) and static representation of objects i and k (slots). The output is added to update the dynamic representation from the dynamic distillation module. Ultimately, the future state is predicted using a linear transformation of the static representation of time t and the updated dynamic representation. \
The paper evaluates their learned representation on the CLEVRER dataset for video reasoning, prediction, reconstruction, and segmentation.

**Summary Of The Review:**

The modules proposed in this work are well motivated and designed and produce strong results on video reasoning. Yet, I am concerned about the insufficient evaluation. Without the ablations and the baseline I discussed in weaknesses it is difficult to quantify the actual impact of the proposed method and its underlying modules. Moreover, experimental details are missing which makes it difficult to reproduce its reported numbers. That is why my initial rating is below the acceptance threshold. But I am open to adjusting my rating if the authors do sufficiently address the points raised in weaknesses.

---

> ### Author Response · Authors · 2021-11-22
> **Author Reply**
>
> We sincerely thank the reviewer for the review and thoughtful feedback. We respond to the points brought by the reviewer below:
>
> **Weakness 1: Missing ablation experiment for dynamic distillation and relation module...**
>
> **Reply:**  We add an ablation experiment of "ODDN w/o relation".  We report results on both video reasoning (Table 2) and video prediction (Figure 3, Figure 7) tasks. The relation module is built on the dynamic distillation module, and when ODDN reduces both dynamic distillation and relation modules ODDN becomes standard IODINE which is already reported.
> - For the reasoning task, "ODDN w/o relation" is worse than ODDN and better than IODINE which is reasonable because "ODDN w/o relation" is "IODINE + dynamic distillation" and ODDN is "IODINE + dynamic distillation + relation". We further notice that "ODDN w/o relation$^{-dyn}$" is a little bit worse than IODINE, which proves that: (1) The distilled dynamics is significant (2)"ODDN w/o relation" does not improve static representation quality as ODDN does. We believe this experiment verifies the impact of both modules.
> - For the prediction task, as Figure 3 shown, "ODDN w/o relation" can learn object-level dynamics and generate reasonable results for 2
> frames. Then when the collision happens, objects blend together and the following frames become blurry.
>
> **Weakness 2: Missing baseline...**
>
> **Reply:**
> - We implement the baseline as suggested: replacing the dynamics distillation module and relation module with a feed-forward network("IODINE linear". The prediction results are shown in Figure 3 and Figure 7: The model does not converge well, the predicted frames seem to be identical.
> - We would like to explain what "$^{-dyn}$"("w/o dyn") means: We train standard ODDN to obtain static representations (dim=16) and distilled dynamics (dim =4), then we reduce the dynamics and input 16 dimension static representations to ALOE to verify whether the static representations are better than IODINE, thus we do not only use static representations to train ODDN (predict the future).
>
> **Weakness 3: The paper compares their method to PROVIDE which is the only other competitor that leverages temporal information...**
>
> **Reply:** We use the released source code to implement PROVIDE. By comparing the training procedure of IODINE and PROVIDE, PROVIDE converge way faster than IODINE. Actually, the reproduced result of PROVIDE is better than the PROVIDE paper (MSE 0.000148 vs 0.0003), we looked up the training details noticed that we use RTX 3090 (24G) to implement all experiments and origin PROVIDE is implemented with RTX 1080 Ti (11G), which leads to the batch size variation. In the PROVIDE paper, the MSE of IODINE is 0.0004, however, in our experiment, IODINE achieves 0.00010 and ODDN with 0.000075. It seems that IODINE-based models need big batch size to fully converge. We did not run experiments in a smaller batch size to verify this because of the time limitation, we will do this in the future. The different performances in reconstruction are positively correlated to the results in Table 2. Unlike ODDN, PROVIDE and IODINE occasionally reconstruct objects with wrong properties (e.g., the gray object in the third raw second column of Figure 4, wrong shape: cube->cylinder), and this is reflected in the representation level. For example, if one question option in CLEVRER be: "The red cube will collide with the sphere“, the reasoning model (ALOE) can't answer it right if the representation is not clear. This comparison gives us extra insights: (1) The reconstruction quality is important because it reflects the representation quality. (2) The representation quality is significant on reasoning task as CLEVRER, at least for ALOE architecture.
>
> **Weakness 4: Missing implementation details**
>
> **Reply:** We add implementation details, hyperparameters, and training procedures to the appendix. The code will be released.
>
> **Weakness 5: The paper evaluates their proposed model only on the CLEVRER dataset.**
>
> **Reply:** ODDN distills and predicts object dynamics for scenes containing object motions consistent with regular physical motion. We don't consider CATER as a proper task to evaluate our method since CATER does not contain natural dynamics and single object dynamics is not the key point of solving this task. However, we conduct experiments on another video dataset: "real block towers". Each video contains a block tower that may or may not be falling. Results prove that ODDN performs better in reconstruction and prediction, details are shown in Figure 6 and Figure 8.

---

> ### Author Response · Authors · 2021-11-22
> **Author Reply P2**
>
> **Weakness 6, 7: Missing related work. Missing comparison to current non-multi object representation-based approaches for video prediction.**
>
> **Reply:** We briefly introduce SRVP in Section 4 **baselines**. We implement SRVP on both CLEVRER and real block towers, comparison results are shown in Figure 3, Figure 7, and Figure 8. On CLEVRER, SRVP seems to fail to capture the tiny motion of a single object, and the predicted future frames are basically identical. For real block towers, SRVP is able to learn object dynamics, however, individuals from predictions generated by SRVP become formless over time.
>
> **Weakness 8: This work does not deal with the inherent ambiguity of the video prediction task itself.**
>
> **Reply:** ODDN is built to learn object dynamics with regular motion (e.g., obeying newton's law), thus ODDN generates deterministic predictions. We point out this limitation of our work in Section 5.
>
> -----
> **We sincerely thank all the general notes**
> - We basically took all the given advice and revised our paper.
> - Figure 7 is now Figure 10 in the revised version. In the dynamics distillation phase, the attention mechanism of the Transformer helps to math objects of two frames. We checked the attention score and the matching was done almost perfectly: attention scores between corresponding objects are close to 1 (0.99 in Figure 10) and others are close to 0.

---

> > ### Comment · Reviewer_YU79 · 2021-11-29
> > **Reponse to Author Replies**
> >
> > Thank you for addressing all my raised concerns in the review. The requested and incorporated ablations and experiments strengthen the paper which is why I adjusted my recommendation in my initial review.

---

### Official Review · Reviewer_pDGg · 2021-11-02

**Correctness:** 2
**Technical Novelty And Significance:** 3
**Empirical Novelty And Significance:** 2
**Recommendation:** 5
**Confidence:** 3

**Main Review:**

The proposed method is compared to the previous methods, IODINE and PROVIDE, and the better results are shown. But, the problem is that the detail of the method is not described and difficult to understand.

In th experiment of video understanding, the input for IODINE is a single frame, but that for the proposed method is multiple frame from video. How different the input data is? If the information extracted by the proposed method is the dynamics of a scene, the importance will depends on the scene. But, no example is shown in the paper to confirm the effectiveness of Dynamic Distillation. Although the method, PROVIDE, is chosen as a method that introduces temporal information, it may not be meaningful because the performance is worse than the base method, IODINE, which uses a single image.

In the experiment of prediction, it is difficult to recognize how is the motion of objects and interaction each other in Fig.4. Is the example appropriate to show if the proposed method can extract the dynamics from video?

In decomposition, what is the condition of the experiment? Is the reconstruction through the autoencoder, or reconstruction from the latent vectors given by user? If in the latter case, what and how are the input latent vectors given?

In the explanation of the proposed method, the indexing of the latent vector is ambiguous. A latent vector is indexed by the frame number i, latent index k, time t and the pixel coordinate. But, what is the difference between the frame number and the time? And the indexes are omitted in many parts of the explanation. It is difficult to understand which latent variables are considered to find the relationship.

The detail of Generative model is not explained. The color is estimated for each pixel. But, the relationship between a latent vector and a pixel is not explained.


**Summary Of The Paper:**

This paper proposes a method of extracting the latent space that describes objects in a video. The encoder part, Inference model, encodes an input image frame-by-frame from a video based on IODINE. The relative information between the latent vectors of three consequent frames is extracted and updated based on Transformer. The input image is reconstructed from the latent vectors by the decoder part, Generative model. The proposed method is tested for various tasks on CLEVRER dataset.


**Summary Of The Review:**

Although the proposed method seems to show the better results in the experiments, the explanation is not sufficient to convince if the experiment is appropriate and fair. Additionally, the explanation of the method is not clear to understand the detail. Additionally, it is not clear that what kind of information is extracted by the part, Dynamic Distillation, introduced in this paper. It seems no experiment is shown that object dynamics is distilled by the proposed method. Consequently, the reviewer thinks that this paper is below the acceptance.

---

> ### Author Response · Authors · 2021-11-21
> **Author Reply**
>
> We sincerely thank you for your efforts and feedback on our work.
>
> **Concern 1:** *In th experiment of video understanding, the input for IODINE is a single frame, but that for the proposed method is multiple frame from video. How different the input data is? If the information extracted by the proposed method is the dynamics of a scene, the importance will depends on the scene. But, no example is shown in the paper to confirm the effectiveness of Dynamic Distillation. Although the method, PROVIDE, is chosen as a method that introduces temporal information, it may not be meaningful because the performance is worse than the base method, IODINE, which uses a single image.*
>
> **Reply:**
> - Firstly, IODINE is originally proposed for static scenes whose input is a single image. We extend to the task of video understanding. In experiments of this video task, the input for all models (ODDN, PROVIDE, and IODINE) is a video. The input is ${x_1...x_N}$ and the output is $z_{11}...z_{1K}$,$...$,$z_{N1}...z_{NK}$, where K is the slot number for each image and each slot corresponding to one object. In another word, $N$ images are inferred into $N \times K$ latent features to represent a video. Then, we use ALOE to handle the reasoning task. Formally, ALOE takes as input the $N \times K$ visual latent features and word embedding sequences of question and choice, then output the answer. We only feed different visual features to ALOE to verify the representation quality.
> - Secondly, we show the effectiveness of Dynamic Distillation in two folds:
>    - (1) In Table 2, we conduct ablations on video reasoning task of with and without object dynamics (ODDN vs ODDN$^{-dyn}$) and  (ODDN w/o relation vs ODDN w/o relation $^{-dyn}$) ), results show the significance of the distilled dynamic representation.
>    - (2) With dynamic distillation and the proposed relation module, ODDN shows better prediction capability compared to other methods (PROVIDE, SRVP).
> - Finally, the reason we compare ODDN with PROVIDE is that both methods encode object dynamics, however, ODDN performs better in representation, prediction, and decomposition. This proves that ODDN should be a better framework to process video inputs for multi-object scenes.
>
> **Concern 2:** *In the experiment of prediction, it is difficult to recognize how is the motion of objects and interaction each other in Fig.4. Is the example appropriate to show if the proposed method can extract the dynamics from video?*
>
> **Reply:** Figure 4 refers to Figure 3 in the revised version. We add markers on the colliding objects (the
> blue and the brown cylinder). Both ODDN and PROVIDE can predict the blue cylinder is moving from left to right, however, PROVIDE can not handle collision and the prediction quality is poor.
>
> **Concern 3:** *In decomposition, what is the condition of the experiment? Is the reconstruction through the autoencoder, or reconstruction from the latent vectors given by user? If in the latter case, what and how are the input latent vectors given?*
>
> **Reply:** We reconstruct the image through the autoencoder, and the input latent vectors are provided by the inference module of different models (IODINE, PROVIDE, and ODDN).  IODINE could decompose the scene into object-centric latent vectors and unsupervised segmentation masks based on amortized iterative refinement and reconstruction self-supervision. We built ODDN based on IODINE by adding object distillation module and relation module shown in Figure 2. By decomposition, we evaluate the segmentation and reconstruction quality. Both PROVIDE and ODDN use temporal clues and obtain better segmentation masks. However, PROVIDE get poor reconstruction results while our ODDN achieves finer reconstruction than IODINE.
>
> **Concern 4:** *In the explanation of the proposed method, the indexing of the latent vector is ambiguous. A latent vector is indexed by the frame number i, latent index k, time t and the pixel coordinate. But, what is the difference between the frame number and the time? And the indexes are omitted in many parts of the explanation. It is difficult to understand which latent variables are considered to find the relationship. The detail of Generative model is not explained. The color is estimated for each pixel. But, the relationship between a latent vector and a pixel is not explained.*
>
> **Reply:** The frame index is the same as the timestep. We now use N to represent timestep/frame and one latent could be fully described as $z_{nk}^t$. A latent vector can not be indexed by the pixel coordinate, the relation between pixel and latent is: all K latents
> are inferred from pixels and each latent could be decoded to pixels and masks. The generative model is the same as IODINE which takes
> one latent as input and expands the latent to $H \times W$, then decodes to a $H \times W \times C$ image and corresponding mask with several convolutional layers.

---

> > ### Comment · Reviewer_pDGg · 2021-11-30
> > **Update**
> >
> > After reading the author response and the revised manuscript, the concerns of the reviewer, the relationship with the base method, IODINE, and the contribution of the proposed method, are made clear.
> >
> > It will be better to explain the structure of the Generative model, including the input and output, even if it is the same with IODINE.

---

> > > ### Author Response · Authors · 2021-11-30
> > > **Response to reviewer update**
> > >
> > > Dear Reviewer,
> > >
> > > Thank you for your supportive review of the revised version.  Now we describe the Generative model, input, and output in detail:
> > >
> > > - For the input and output part:
> > > ODDN takes a video as input, and the output:
> > >   - Reconstructed images for each frame of the video (Based on inferred latents from the  Inference model).
> > >   - Predict (n+2)th frame for each n($1...(N-2)$) (Based on predicted latents from the Relation model).
> > >
> > > - For the generative part:
> > >   - Each object latent $z_{nk}$ is decoded separately into pixel-wise means $\mu_{nk}$ and mask-logits $\hat{m_{nk}}$, which we then normalize using a softmax operation applied across slots such that the masks  $\hat{m_{nk}}$ for each pixel sum to 1. Together, $\mu$ and $m$ parameterize the spatial mixture distribution as defined in Equation (1).
> > >
> > >   - For the network architecture of the Generative model, we use a broadcast decoder (Watters et al., 2019), which spatially replicates the latent vector $z_{nk}$, appends two coordinate channels (ranging from −1 to 1 horizontally and vertically), and applies a series of size-preserving convolutional layers. This structure encourages disentangling the position across the image from other features such as color or texture and generally supports disentangling. All slots k share weights to ensure a common format and are independently decoded, up until the mask normalization.
> > >
> > > We will take your advice and explain the  Generative model, input, and output in the final version of our paper.
> > >
> > > [1] Watters et al., 2019. "Spatial broadcast decoder: A simple architecture for learning disentangled representations in vaes."

---

### Official Review · Reviewer_dfem · 2021-11-02

**Correctness:** 4
**Technical Novelty And Significance:** 3
**Empirical Novelty And Significance:** 3
**Recommendation:** 8
**Confidence:** 3

**Main Review:**

Strengths:

- The paper presents a very interesting idea of using transformer structure to align latent features and distillate object dynamics, and it seems very effective.

- Experimental results show superior performance compared to the SOTA on video understanding and reasoning.

Weaknesses:

- The paper is built upon many existing techniques, including the IODINE, NPE, and R-NEM. Especially, the core part of it is the use of IODINE latents and aligning them by a transformer structure. The reasoning module is a combination of NPE and R-NEM.


**Summary Of The Paper:**

The paper presents a framework that distillates explicit object dynamics (e.g., velocity) from the discrepancies between objects’ static latents of different frames, called the Object Dynamics Distillation Network (ODDN). The approach is built upon recent works which decompose static scenes into independent latents by randomly assigning different objects to a fixed number of slots, which share weights to obtain latents with a common format. And this paper makes use of the attention mechanism of the transformer to align objects’ latents of two input images and encode the aggregated representation into a low-dimensional vector to obtain disentangled object dynamics. Afterward, the paper builds a relation module to model object interactions based on NPE and R-NEM.

**Summary Of The Review:**

Though the paper is built upon many existing techniques, including the IODINE, NPE and R-NEM, and the novel part of this paper is using a transformer structure to align the IODINE latents and find discrepancies, I still consider this paper very interesting and strong. It may provide some insight on how to model individual object dynamics from videos. Moreover, the proposed method is very effective as experimental results show superior performance.

---

> ### Author Response · Authors · 2021-11-21
> **Author Reply**
>
> We sincerely thank the reviewer for the comments and the recognition of our efforts.
>
> **Weakness:** *The paper is built upon many existing techniques, including the IODINE, NPE, and R-NEM. Especially, the core part of it is the use of IODINE latents and aligning them by a transformer structure. The reasoning module is a combination of NPE and R-NEM.*
>
> **Reply:**
> Our method is based on NPE and R-NEM, however, to our knowledge, ODDN is the first to explicitly and successfully model object interactions on CLEVRER. The relation module endows the model to predict reasonable future frames iteratively and helps the inference model to generate higher quality latent features.

---

> > ### Comment · Reviewer_dfem · 2021-12-01
> > **Post Rebuttal**
> >
> > I leaned toward 'acceptance' of this paper initially. Though I consider the paper borrowed much technique from existing work, the idea of using transformer structure to align latent features and distillate object dynamics is novel and has merit. Since the feedback has resolved many concerns about the experiments raised by other reviewers, I will stick to my original score. Thanks

---

### Official Review · Reviewer_icxn · 2021-11-03

**Correctness:** 4
**Technical Novelty And Significance:** 3
**Empirical Novelty And Significance:** 3
**Recommendation:** 6
**Confidence:** 3

**Main Review:**

[Strengths]
The authors provide a novel distillation method to understand object dynamics. The proposed system shows state-of-the-art performance on CLEVRER.

[Weaknesses]
- I would like to see more qualitative results on the attention module. Visualized coefficients in Fig. 6 only cover one single scenario. I think the paper needs more validations on whether the attention module converges well.

- The proposed method is validated only in CLEVRER, which does not show its generalization capability. In addition, it is hard to find the differences in the qualitative results (Fig. 4) compared to the previous methods. In Fig. 5, IODINE cannot represent the shadow areas, while the proposed ODDN can reconstruct them. Could you elaborate on the reason how ODDN has a representation ability on the shadow areas?

- I would like to see the comparison of the computational complexity with the previous methods, e.g., ALOE, IODINE, etc.

- Please discuss the limitations of the proposed method. It is required to discuss whether there are any other aspects that can be further improved (as future works) for this task.


**Summary Of The Paper:**

This paper presents an unsupervised distillation of disentangled object dynamics from raw video inputs. The distilled dynamics model is capable of causal reasoning and future frame prediction. Extensive results on tasks of segmentation and reconstruction show its favorable performance.

**Summary Of The Review:**

Overall, the proposed architecture is designed simple and shows state-of-the-art performance. There are minor issues as commented in [Main Review], but I am leaning towards positive at this moment. Since the topic is out of my scope, I would like to see other reviewers' opinions.

---

> ### Author Response · Authors · 2021-11-21
> **Author Reply**
>
> We sincerely thank the reviewer for the helpful comments and suggestions.
>
> **Weakness 1:** *I would like to see more qualitative results on the attention module. Visualized coefficients in Fig. 6 only cover one single scenario. I think the paper needs more validations on whether the attention module converges well.*
>
> **Reply:** We removed Figure 6 and add an additional example and more detailed visualization to Figure 9 (appendix), which shows attention coefficients the attention module learns over 7 timesteps, and a collision happens. We can see the attention module can attend to the approaching object when the collision is about to happen (the rubber sphere and the metal sphere).
>
> **Weakness 2:** *The proposed method is validated only in CLEVRER, which does not show its generalization capability. In addition, it is hard to find the differences in the qualitative results (Fig. 4) compared to the previous methods. In Fig. 5, IODINE cannot represent the shadow areas, while the proposed ODDN can reconstruct them. Could you elaborate on the reason how ODDN has a representation ability on the shadow areas?*
>
> **Reply:**
> - Firstly, we conduct experiments on another dataset: "real block towers", where each video contains a block tower that may or
> may not be falling. The Results prove that ODDN performs better in reconstruction and prediction, details are shown in Figure 6 and Figure 8 of the revised paper.
> - Secondly, in Figure 3, which is Figure 4 of the original version, we add markers to highlight the collision of the blue and brown cylinder, and add the captions to highlight the difference: "In the video, a collision happens between the blue and brown cylinder from timestep 6 to 8." ODDN could output reasonable predictions and handle objects collision. PROVIDE could only predict an object’s future state independently. The reason is that PROVIDE does not encode the object's interaction, and objects become blended when a collision is supposed to happen and object dynamics change.
> - Finally, The ability to reconstruct shadow areas of ODDN is endowed from the relation model. The relation model computes whether close objects (usually come along with shadow areas) collide with each other or not, thus more appearance texture is captured, which reconstructs better appearance when collision may happen.
>
> **Weakness 3:** *I would like to see the comparison of the computational complexity with the previous methods, e.g., ALOE, IODINE, etc.*
>
> **Reply:** We add computational complexity comparison to the appendix in Table 4. We compare GFLOPs, parameters, and model size.
> Since ALOE is designed for downstream task: video reasoning, we only compare ODDN with PROVIDE and IODINE.
>
> **Weakness 4:** *Please discuss the limitations of the proposed method. It is required to discuss whether there are any other aspects that can be further improved (as future works) for this task.*
>
> **Reply:** We add limitation discussion and future direction to section 5: DISCUSSION AND FUTURE WORK:
>
>  *ODDN also has limitations. The prediction quality decrease over time because of the error accumulation. ODDN can only generate deterministic predictions for dynamics obeying regular patterns.
> Besides, ODDN needs two frames to firstly derive object dynamics then predict the future. However,
> in the block tower case, humans can tell whether the tower will fall or stay stable with one frame by
> commonsense. We leave this for future work.*

---

> > ### Comment · Reviewer_icxn · 2021-11-29
> > **Post-rebuttal**
> >
> > I appreciate the authors' feedback and valuable comments from other reviewers. The authors' feedback supports their novelty and resolves the weaknesses. Overall, I would like to keep my score.

---

### Official Review · Reviewer_xJFb · 2021-11-03

**Correctness:** 3
**Technical Novelty And Significance:** 2
**Empirical Novelty And Significance:** 3
**Recommendation:** 6
**Confidence:** 2

**Main Review:**

# Strengths

### Novelty
- The two key contributions are clearly described:
    - use explicit dynamic features learnt by matching objects’ latents between the current and previous frames as opposed to only learning static scene properties as done in prior work.
    - explicitly model interactions between objects

### Experimental verification
- Small ablation on the effect of dynamic features in Table 2
	 - Even if dynamics features are absent the model achieves better results than IODINE which means that ODDN can both model object dynamics and generates overall higher quality object static representations.

- Gives SoTa improvement across a variety of tasks
    - Table 1 and Table 2 show a consistently significant improvement over prior work
    -  ODDN Model representations are fed into ALOE instead of using static Monet features which ALOE originally used. This ODDN-ALOE model improves on state of the art on CLEVER for various tasks
    - Interestingly the gains are greatest on predictive and counterfactual questions which are related to predicting how objects will move.
    -  ODDN also improves over using object representations from IODINE or PROVIDE (in Table 2).
    - Qualitatively, in Fig 4, the generated images for the next frames are better for ODDN as it is able to preserve the color of the objects as well as position whilst PROVIDE fails to preserve color after a few frames
    - ODDN also does better at reconstructing images and generates more compact  segmentation masks than IODINE or PROVIDE (in Fig 2).

# Weaknesses

### Clarity of explanations and figures could be greatly improved
- Although the ideas are clear, the method and implementation could be made clearer e.g. write out the algorithm in pseudo code
    - In addition, the notation is slightly confusing given there are two meanings for t - inference model time step and time step in the video
- Fig 7 (c) + section titled “Disentanglement” is not clearly explained
    - The caption is incorrect - (b) is actually referring to (c). (a) is not explained.
    - What is a dynamic dimension?
    - You could add some arrows to the diagram to explain what is happening across the rows/columns. It’s not clear how to interpret the images.
    - In addition where does “velocity magnitude” come from? This was not explained previously.
- Figure 4 is also not very clear
    - Explain in the caption what is happening in the 6 frames i.e. blue and brown cylinder collide, with ODNN this interaction is predicted but with PROVIDE it loses information about the blue cyclinder. It would be useful to explicitly mention the colors to look for in the images in the caption or in Section 4 in the text.
    - The images are also really small - hard to see what is happening
- Figure 3
    - Is it showing MSE from 6 frames in total or averaged over some number of runs?
- Table 3
    - For each metric add an arrow to indicate if lower or higher is better
- Figure 6 is also not clear
    - The caption indicating the colors for the frames and timestamps is really tiny

### Lack of implementation details
- No details about how the model was implemented or how to reproduce the results
- Equation 2 and 3 explain what IODINE does but the ODDN method is not fully explained.

### Typos:
- Section 4.3: “weather” -> “whether”


**Summary Of The Paper:**

The paper presents a method which takes as input sequences of video frames of scenes from which it is able to understand the dynamics of objects present in the scene and their interactions and transfer this to several downstream tasks. The model consists of two modules, one to distill individual object dynamics and a second relation module to understand interactions between objects.  The model is shown to have SoTA results on several downstream tasks including video understanding and reasoning, video prediction, reconstruction, and segmentation. The representations learnt by the model are shown improve upon prior work especially on tasks which require knowing object dynamics such as predicting the next frame in a sequence where collisions or occlusions can occur.

**Summary Of The Review:**

The paper presents a method with two key insights/contributions that are not present in prior work. The ideas are validated quantitatively and qualitatively. The technical details, figures and tables could be improved with better explanations and descriptions.

---

> ### Author Response · Authors · 2021-11-21
> **Author Reply**
>
> First of all, we thank the reviewer for the detailed comments and reviews.
>
> **Concern 1:** *Although the ideas are clear, the method and implementation could be made clearer e.g. write out the algorithm in pseudo-code. In addition, the notation is slightly confusing given there are two meanings for t - inference model time step and time step in the video.*
>
> **Reply:** We add pseudo code to appendix A.1.  We now use $N$ to represent input frame timesteps,  $T$ to represent inference timesteps, and $K$ to represent slots number. One latent feature could be written as $z_{nk}^t$
>
> **Concern 2:**  *Fig 7 (c) + section titled “Disentanglement” is not clearly explained*
>
> **Reply:**  Now we put Fig. 7 into the appendix as Fig. 10
> - We correct the assignment of the (a),(b),(c) indexes and their corresponding captions.
> - ODDN distills object dynamics into a low dimensional vector (4 dimensions in the picture), where "one dynamic dimension" is referred as one dimension of the feature vectors. We have revised it to "the corresponding dimension of the dynamic representation" for clearer presentation.
> - We add markers to better interpret what is going on.
> - The "velocity magnitude” refers to the dynamic factor value that controls how fast the object location changes. We delete it in the new version.
>
>
> **Concern 3:** *Figure 4 is also not very clear*
>
> **Reply:** Now Figure 4 refers to Figure 3 in the new version. We add extra captions to describe the event in the image sequence, and how the result becomes better by the proposed components. We add makers to capture the key areas
>
> **Concern 4:** *Figure 3. Is it showing MSE from 6 frames in total or averaged over some number of runs?*
>
> **Reply:** Now Figure 3 refers to  Figure 7 in the new version.  It shows the average MSE of each of the predicted 6 frames over the entire test set (1000+ videos)
>
> **Concern 5:** *Table 3. For each metric add an arrow to indicate if lower or higher is better*
>
> **Reply:** We add arrows to indicate if lower or higher is better
>
> **Concern 6:** *Figure 6. The caption indicating the colors for the frames and timestamps is really tiny*
>
> **Reply:** We removed Figure 6. Detailed attention visualization is shown in Figure 9.
>
> **Concern 7:** *No details about how the model was implemented or how to reproduce the results*
>
> **Reply:**  We add implementation details, hyparameters, and training procedures into the appendix
>
> **Concern 8:** *Equation 2 and 3 explain what IODINE does but the ODDN method is not fully explained.*
>
> **Reply:**  ODDN is built on IODINE. Basically, IODINE is an auto-encoder framework, and based on the latent features IODINE inferred, ODDN distills object dynamics (Equation 8 and 9) and models their relations then predicts the next frame latent representations (Equation 4 and 6).

---

> > ### Comment · Reviewer_xJFb · 2021-11-23
> > **Update**
> >
> > Thanks for addressing my comments in detail and incorporating the suggestions to improve the clarity of the paper.

---

### Decision · Program_Chairs · 2022-01-20

**Decision:**

Accept (Poster)

**Comment:**

This work presents a novel method h to learn object dynamics from unlabelled videos and shows its benefits on causal reasoning and future frame prediction.  This paper received 4 positive reviews and 1 negative review. In the rebuttal, the authors have addressed most of the concerns. AC feels this work is very interesting and deserves to be published on ICLR 2022. The reviewers did raise some valuable concerns that should be addressed in the final camera-ready version of the paper. The authors are encouraged to make other necessary changes.